# Enhanced Tracer Particle Detection in Dynamic Bulk Systems Based on Polarimetric Radar Signature Correlation

**DOI:** 10.3390/s24092673

**Published:** 2024-04-23

**Authors:** Birk Hattenhorst, Nicholas Karsch, Thomas Musch

**Affiliations:** Institute of Electronic Circuits, Ruhr University Bochum, Universitätsstr. 150, 44801 Bochum, Germany

**Keywords:** tracking, radar, polarimetry, particulate system, tracer particle

## Abstract

This contribution focuses on the detection of tracer particles within non-homogeneous bulk media, aiming to enhance insights into particulate systems. Polarimetric radar measurements are employed, utilizing cross-polarizing channels in order to mitigate interference from bulk media reflections. To distinguish the tracer particle in the measurements, a resonant cross-polarizing structure is constructed, facilitating the isolation of frequency signatures from the surrounding bulk clutter. In addition to characterizing the bulk and tracer components, this study provides a detailed presentation and discussion of the measurement setup, along with the employed signal processing methods. The effectiveness of the proposed methods is demonstrated through comprehensive measurements, where a tracer particle is systematically positioned at various locations. The results affirm the feasibility and efficacy of the approach, highlighting its applicability for enhanced dynamic monitoring in particulate systems within industrial processes.

## 1. Introduction

In particulate system research, major areas of interest are the chemical and physical processes in moving and reacting fluidized granular assemblies, especially bulks. The reason for this is the strong existence of particulate systems in industrial sectors. These include chemistry, energy, food, and pharmaceutics, which are indispensable for industrialized countries. Besides their omnipresence in industry and nature, a complete understanding of the interaction and movement of granular particles is still not achieved due to the non-trivial description as a continuum. To overcome this challenge, computer simulations greatly evolved and enabled predictions of movement patterns in bulks. Despite this, difficulties arise when simulating single particle movements or no suitable verification can be done. As a result, diverse measurement technologies have been developed and applied to gain further insight into particulate systems. These technologies can be grouped into imaging and tracking systems [1] with their inherent advantages and disadvantages.

In the case of imaging, magnetic resonance imaging (MRI) [2], electrical capacitance tomography (ECT) [3], and Röntgen radiation (X-ray) tomography [4] are widely used. Their ability to provide fully three-dimensional images with high spatial resolution of the particulate system under testing comes with the downside of low measurement rates. Furthermore, these technologies are limited by scalability and particle type, and are typically very costly. For these reasons, they are impractical for large-scale industrial processes. Other imaging techniques rely on optical approaches, such as digital cameras [5] and confocal microscopy [6]. They are highly dependent on the opacity of granular particles. For non-transparent particles, the applicability of these methods is reduced to two-dimensional measurements. Moreover, only relatively small areas can be monitored. These reasons restrict the optical methods.

Tracking methods typically require a traceable particle that differs from the surrounding granular particles in at least one of its physical properties, such as radioactivity or magnetic dipole moment. Simultaneously, the tracer’s mechanical properties—such as mass, size, damping, and friction—should ideally mirror those of the particles in the process. Established methods are mainly positron emission particle tracking (PEPT) [7], radioactive particle tracking (RPT) [8], and magnetic particle tracking (MPT) [9]. The downsides of these measurement techniques lie in the tracer particles themselves, which are either radioactive (and, therefore, not secure) or relatively heavy due to the magnetic materials used. MPT, however, benefits from the directional magnetic field of the magnetic dipole, which allows for tracking of the position and orientation simultaneously.

In addition to the presented methods, electromagnetic measurement techniques are very promising in the field of moving bulk and granular research. Similar to the established methods, electromagnetic measurement techniques can be grouped into imaging and tracking systems. Microwave tomography (MWT) [10,11] has demonstrated its imaging capabilities in industrial flow and granular processes. Nevertheless, due to the tomographic approach, relatively small measurement cells can be monitored in a two-dimensional plane. Radar-based approaches like synthetic aperture radar (SAR) imaging [12,13] and polarimetric ground penetrating radar (GPR) [14,15,16] are widely used in nondestructive testing applications. Here, usually, the object under test is stationary so that a synthetic antenna aperture can be generated by moving the measurement antenna and carrying out multiple measurements. Along with high computational expenses, this limits the usage of real-time measurements of moving processes. Also, the surrounding media in the previously mentioned works are significantly finer-grained than the medium used in this work.

For tracking, radar measurements offer great potential to overcome the limitations of established tracking techniques. Here, typically, array or multiple-input multiple-output (MIMO) configurations (MIMO) [17] are used to track individual particles. In [18,19], a radar-based three-dimensional array was utilized to track corner reflectors embedded in particles as tracers. The accurate non-invasive measurement procedure with high measurement rates and its scalability to large-sized industrial setups are advantages of the radar-based approach. However, in this case, the designed tracers only permit monostatic measurements due to their working principle. Moreover, the rather large tracer sizes that are needed to work properly are a downside.

Lately, a MIMO frequency-modulated continuous-wave (FMCW) radar approach alongside a metallic sphere particle as a tracer has been used [20]. In Figure 1, the principle measurement setup is illustrated.

Here, several (as little as possible) antennas are positioned around a reactor in their respective far fields. The reactor with dielectric windows contains non-conducting packed particles in different shapes and sizes. Frequencies from 1.5 to 8.5 GHz present a good compromise between the detectability of the tracers and clutter reduction from the surrounding bulk particles. In the aimed application, bulk spheres with a diameter in the order of centimeters are present in the reactor. As for all tracking techniques, the tracers should have equivalent mechanical properties compared to the surrounding bulk particles to ensure matching dynamic behavior. Under these conditions, embedded corner reflectors or metalized tracers can be detected as long as they are sufficiently large and possess larger radar cross-sections (RCSs) than the surrounding particles. Nevertheless, occasional malfunctions can occur, and complete failure is possible when their physical cross-sections align with those of the bulk particles. Additionally, they operate with the disadvantage of being limited to monostatic setups or sharing the same major polarization as the cluttering bulk. In this contribution, a polarimetric FMCW radar approach is introduced that utilizes polarizing tracers. They distinguish themselves from the aforementioned established tracers due to their cross-polarization, which introduces measurement effects in polarimetric measurement channels of the system where the bulk exhibits less clutter. This is achieved by embedding resonant structures like dipoles into dielectric particles. Therefore, the proposed approach is also scalable.

This work is structured as follows: Section 2 lays the theoretical foundation for the developed measurement principle, Section 3 presents the electromagnetic scattering behaviors of different particle assemblies and tracers, Section 4 reviews the used setup and methods, Section 5 discusses the measurement results, and Section 6 concludes the main findings of this work.

## 2. Fundamentals

### 2.1. Radar Cross-Section

The complex RCS σ of an arbitrary object in the scattering direction $\left(\theta_{s},\varphi_{s}\right)$ for a small solid angle $d\Omega_{s}$ refers to [21], defined by the radiated electric field vector ${\vec{E}}_{r}$ at distance *R* in relation to the incident field ${\vec{E}}_{i}$ at the object coming from the direction $\left(\theta_{i},\varphi_{i}\right)$
(1)$$\sigma \left(R,\theta_{s},\varphi_{s},\theta_{i},\varphi_{i}\right)=\lim_{R\to \infty }4\pi R^{2}{\left(\frac{|{\vec{E}}_{r}\left(R,\theta_{s},\varphi_{s}\right)|}{|{\vec{E}}_{i}\left(\theta_{i},\varphi_{i}\right)|}\right)}^{2}\exp\left[j2\beta (R-r)\right].$$ In (Equation 1), β is the phase constant of the electromagnetic wave and *r* is the radius of a sphere around the object, as shown in Figure 2.

Moreover, φ and θ are the azimuth and elevation angles for the incident (i) and scattered (s) wave, respectively. As an assumption for (Equation 1), the point of observation or the receiving antenna has to be in the far field of the scattering object. Further, the object itself has to be in the far field of the emitting antenna. Practically, a distance of R→∞ cannot be achieved. Therefore, the conventional far-field definition assumes a distance expressed as follows: (2)$$R\geq \frac{2D^{2}}{\lambda },$$
referring to [22]. In (Equation 2), *D* is the maximum dimension of either the object under test or the emitting/receiving antenna. As an assumption for (Equation 2), the dimensions of the object or antenna are larger than the wavelength (D>>λ). In case the scattering object under test is small compared to the wavelength, a second far-field condition needs to be satisfied
(3)$$R\geq \frac{\lambda }{2\pi }.$$

For both conditions, Figure 3 exemplarily displays the necessary distances for different particle and antenna sizes with respect to the actual measurement setup that will be described in Section 4.

A drawback of (Equation 1) is the distance dependency of the amplitude and phase. To compensate for this, the RCS is referenced to as the plane at *r* by
(4)$$\begin{array}{cc} \sigma (\theta_{s},\varphi_{s},\theta_{i},\varphi_{i}) & =4\pi r^{2}{\left(\frac{|{\vec{E}}_{s}\left(r,\theta_{s},\varphi_{s}\right)|}{|{\vec{E}}_{i}\left(\theta_{i},\varphi_{i}\right)|}\right)}^{2}\exp\left[j2\left(\phi_{s}-\phi_{i}\right)\right] \end{array}$$
where $\phi_{s}$ and $\phi_{i}$ are the phases of the incident and the scattered field. In addition, due to the possible depolarization mechanisms of the tracer and the granular particles, polarization has to be considered in the previously mentioned quantities.

### 2.2. Polarimetry

Polarimetry is a technique that uses polarized electromagnetic waves to study and differentiate the properties of different scatterers. The direction of the electric field of the wave defines its polarization. In a simple monostatic scenario, as for the left antenna in Figure 2, the polarization is vertical (V). In this context, the vertical polarization component is referred to as co-polarization, while the orthogonal horizontal (H) component is known as cross-polarization. However, this classification is not sufficient for the bistatic scenario of two antennas, as in Figure 2. Therefore, multiple polarization systems have been defined in the literature [23,24] for the three-dimensional characterization of scatterers and antennas. In this contribution, the Ludwig II (Azimuth over Elevation) polarization system is applied in all polarization considerations. This is due to the center-focused arrangement of the antennas in Figure 1. Here, the co- and cross-polarization components are oriented along the azimuth and elevation orbits, as can be seen in Figure 2. For simplification, the H and V terms for the monostatic horizontal and vertical polarization components will be used in the following for the orthogonal field components at each antenna.

Quantifying scattering mechanisms starts by encoding the received signal in a scattering matrix S, which connects the transmitted to the received field components [21].
(5)$$\begin{array}{c} \left[\begin{array}{c} E_{H,s}\\ E_{V,s} \end{array}\right]=\left[\begin{array}{c} S \end{array}\right]\left[\begin{array}{c} E_{H,s}\\ E_{V,i} \end{array}\right]=\left[\begin{array}{cc} S_{\mathrm{HH}} & S_{\mathrm{HV}}\\ S_{\mathrm{VH}} & S_{\mathrm{VV}} \end{array}\right]\left[\begin{array}{c} E_{H,i}\\ E_{V,i} \end{array}\right]. \end{array}$$ In (Equation 5), $\left[\begin{array}{c} S \end{array}\right]$ is the object-related scattering matrix that is independent of the distance, *R*, and is related to the RCS of (Equation 4) by
(6)$$\left[\begin{array}{c} S \end{array}\right]=\frac{1}{\sqrt{4\pi r^{2}}}\left[\begin{array}{c} \sqrt{\sigma } \end{array}\right]$$
with
(7)$$\begin{array}{c} \left[\begin{array}{c} \sqrt{\sigma } \end{array}\right]=\left[\begin{array}{cc} \sqrt{\sigma_{\mathrm{HH}}} & \sqrt{\sigma_{\mathrm{HV}}}\\ \sqrt{\sigma_{\mathrm{VH}}} & \sqrt{\sigma_{\mathrm{VV}}} \end{array}\right]. \end{array}$$ To extract physical information regarding scattering objects, it is imperative to investigate the second-order statistics between pairs of polarization channels. To initiate this analysis, the vectorization of [S] is essential and can be described as in [25]: (8)$$\begin{array}{c} k=\frac{1}{2}\mathrm{Trace}\left(\left[S\right]\left[\Phi \right]\right)={\left[\begin{array}{cccc} k_{0} & k_{1} & k_{2} & k_{3} \end{array}\right]}^{T}. \end{array}$$
where [Φ] represents a set of 2 × 2 complex basis matrices. The choice of these basis matrices varies for different applications and is therefore dependent on the type of decomposition method employed to investigate the scattering process. Given this context, the second-order statistics can be examined using the correlation matrix, as follows: (9)$$\left[T\right]=k\cdot k^{*}$$
where ^∗^ denotes the complex conjugate transpose. The correlation matrix is a square and symmetric matrix that comprehensively captures the correlation between all conceivable pairs within the vector k. Larger off-diagonal terms within the matrix indicate redundancy, while smaller off-diagonal terms suggest statistical independence. To distinguish among scattering mechanisms, additional considerations may be given to the eigendecomposition of the correlation matrix, denoted by the following: (10)$$\begin{array}{c} \left[T\right]=\left[V\right]\left[\Lambda \right]{\left[V\right]}^{-1}=\left[\begin{array}{ccc} v_{1} & v_{2} & v_{3} \end{array}\right]\left[\begin{array}{ccc} \lambda_{1} & 0 & 0\\ 0 & \lambda_{2} & 0\\ 0 & 0 & \lambda_{3} \end{array}\right]\left[\begin{array}{c} v_{1}\\ v_{2}\\ v_{3} \end{array}\right]. \end{array}$$ The matrix [V] represents the eigenvectors of [T] organized in columns. Given the symmetry of [T], these eigenvector columns are orthogonal, establishing the relationship ${\left[V\right]}^{-1}=\;{\left[V\right]}^{T}$ [26]. Consequently, the eigenvectors $v_{i}$ characterize the directions associated with the most prominent scattering mechanisms. The orientation of each eigenvector can, thus, serve as a metric for discerning distinct orientation directions of identical scattering objects. Simultaneously, the corresponding eigenvalues $\lambda_{1}>\lambda_{2}>\lambda_{3}>0$ quantify the intensity of these scattering mechanisms.

## 3. Scattering Characterization of Single and Packed Particles

Particle systems exhibit diverse configurations, varying in system and particle size, form, and arrangement. In navigating this diversity, three-dimensional electromagnetic simulations have proven invaluable in the system design process by offering insights into the scattering behavior of the bulk. The key advantage of the simulation approach lies in its independence from a supporting structure, such as a reactor holding particles in place. Consequently, the characterization is not obscured by structural clutter, and particle arrangements are more straightforward to set up compared to practical cases. In the subsequent analysis, we explore the polarimetric scattering behaviors of different bulk configurations in the frequency domain. This exploration aids in identifying frequency regions where designed tracer particles can provide polarimetric features, facilitating their separation from clutter. The simulations were conducted using CST Microwave Studio (Dassault Systèmes, Vélizy-Villacoublay, France) with the time domain solver and a hexahedral mesh. In all simulations, a plane wave excites the environment within the frequency range of 1.5 to 8.5 GHz, propagating in the positive z-axis direction. The polarization or electric field vector is aligned with the y-axis. Over the frequency range, 1001 RCS monitors capture the scattered power in all spherical directions at distinct frequencies. A high accuracy with a steady state energy check of −80 dB was chosen and reached for all simulations due to a high maximum solver duration. These settings account for resonance phenomena like dipole structures or dielectric resonances in the time domain. For simplicity of depiction, the following results display only the xz-plane (elevation angle $\theta =0^{\circ }$), corresponding to the horizontal plane of the antennas in Figure 1.

### 3.1. Single Particle Scattering

The initial focus is on investigating the scattering behavior of single bulk particles. Given the intricate nature of considering all possible particle geometries, this study concentrates on spherical particles, which are expected to exhibit small cross-polarization in their RCS referring to the theory of Mie scattering [27]. To explore this, simulations of a dielectric sphere with a relative permittivity of $\varepsilon_{r}=3.2$, and a diameter of 20 mm, were conducted. The resulting co- and cross-polarized RCSs are illustrated in Figure 4.

In this representation, the spherical axis corresponds to the azimuth angle, with the direction of propagation aligning with the positive z-axis, equivalent to φ=0°. The radial axis represents the frequency axis, providing a second dimension of observation alongside the spatial resolution. The results indicate matching scattering behavior in the co-polarization for the sphere compared with the Mie scattering. Additionally, following Mie scattering principles, the co-polar RCS exhibits permittivity-dependent behavior, as illustrated in [28] and Figure 5. Consequently, higher scattering and cross-polarization are anticipated with increasing permittivity values.

### 3.2. Scattering of Packed Particle Assemblies

Knowing the scattering behavior of a single particle, the composite or aggregate RCS of a particle assembly can be theoretically calculated through the superposition of individual scatterers [29]. However, due to the intricacies and complexities of a randomly packed particle setup, a co-simulation framework is employed involving rigid-body model simulations with the open-source software Blender 3.6 (Blender Foundation) and three-dimensional electromagnetic simulations with CST Microwave Studio. In this approach, the packing of the particle bed is generated in Blender using a rigid-body particle simulation. The use of a non-deformation model is appropriate for robust particles typical in industrial applications and offers the advantage of faster bed packing simulations compared to discrete element method (DEM) simulators. This is facilitated by Blender’s convex hull collision method, which relies on the meshing of the particles [30]. In the simulations, a sufficient surface triangulation was chosen, and beds for three different spherical particle diameters (10 mm, 20 mm, and 30 mm) were generated. The x-, y-, and z-coordinates of all particle centers in the respective beds were exported using a Python script and then imported into CST using a Visual Basic for Applications (VBA) macro. The VBA macro generated dielectric spheres at each particle position. The workflow of the co-simulation framework is depicted in Figure 6.

For a simulation involving a packed bed with dimensions of 300 × 300 × 450 mm³ (width × length × height), Figure 7 illustrates the co- and cross-polarizations in the horizontal plane.

Based on the absolute magnitudes, it is evident that the co-polarization scatters with higher powers compared to the cross-polarization. Additionally, the most pronounced clutter appears in the direction of propagation ($\varphi =0^{{}^{\circ}}$) and in the backscatter case ($\varphi =180^{{}^{\circ}}$) for the co-polarization. This is why, in [20], only channels of bistatic angle were utilized. Regarding cross-polarization, the clutter is evenly distributed over the azimuth orbit. However, clutter diminishes toward lower frequencies, as observed in the radial direction. Overall, the simulations reveal a highly inhomogeneous scattering behavior. Typically, in non- or weakly changing scenarios, clutter reduction is achieved by subtracting a reference measurement from the bulk without a tracer. However, the challenge arises in scenarios with moving particle systems that scatter inhomogeneously, where the reference does not adequately represent the system in subsequent measurements. To address this, another randomly packed particle system was generated and subtracted from the results in Figure 7. The resulting difference between the two simulations is depicted in Figure 8.

In both co- and cross-polarizations, the absolute amplitude of the RCS experienced a very slight reduction. Furthermore, inhomogeneity persisted across both polarizations. This indicates that subtracting a reference from radar measurements of moving particle systems is feasible only in scenarios with minimal changes. Additionally, in a direct comparison of co- and cross-polarizations, the latter exhibits less clutter, making it a more appealing measurement channel. Consequently, the objective of this work is to design tracer particles with strong cross-polarization in frequency regions characterized by minimal clutter, typically found toward lower frequencies, as illustrated in Figure 8b. Moreover, the clutter is evenly distributed along the azimuth, which enables monostatic measurements in cross-polar channels.

The cause of cross-polarization in the bulk remains unidentified. As the spheres themselves exhibit neglectable cross-polarization, the interaction among the spheres is likely the contributing factor. To explore this, two other well-known packings were generated: the simple cubic (SC) packing and the body-centered cubic (BCC) packing. These packings were simulated with the same overall bed volume as the randomized bed. The results and the packing structure are illustrated in Figure 9, with both figures scaled to the same amplitudes as the randomized bed in Figure 7b.

The SC packing achieves cross-polarization levels significantly lower than the randomized bed. Conversely, the BCC packing exhibits indications of cross-polarization. This observation underscores the significant influence of particle packing on cross-polarization, emphasizing the importance of considering packing characteristics in system design.

Finally, it is imperative to investigate the dependencies of the size and relative permittivity of the spheres on the cross-polarization of the packed bed. To explore this, additional packings of spheres with diameters of 10 mm and 30 mm were generated, while keeping the relative permittivity constant at $\varepsilon_{r}=\;3.2$ in both cases. The resulting scattering behaviors are illustrated in Figure 10.

Compared with Figure 7b, it is evident that cross-polarization increases with the sphere diameter, extending into lower frequency regions. For 10 mm spheres, the high clutter region lies outside the considered frequency range. However, regions of moderate clutter still randomly occur, with a higher probability in higher frequency regions. A similar trend is observed for increasing permittivity. To illustrate this, simulations with a randomized bed of 20 mm spheres were conducted for relative permittivity values of $\varepsilon_{r}=\left[2,4,5\right]$, as presented in Figure 11.

Here, the circle of low clutter contracts with higher permittivity values. This study highlights that the cross-polarization clutter generated by randomized packed beds depends on the shape and electrical length of the particles, as well as their packing configuration. Consequently, resolving this clutter through the subtraction of reference measurements poses a considerable challenge.

### 3.3. Tracer Particle Scattering

As mentioned earlier, the core principle in the tracer particle design revolves around generating cross-polarized scattering behavior, particularly in frequency regions characterized by weak clutter resulting from densely packed particles. While subtracting reference measurements is no longer a prerequisite, it can still prove beneficial in enhancing the signal-to-noise ratio (SNR) of measurements. In this study, we propose the use of an embedded crossed dipole within a dielectric sphere as a tracer particle. Crossed dipoles are recognized for their efficacy in polarizing electromagnetic waves during interactions. This attribute arises from the orientation of two dipoles aligned at a 90° angle to each other, enabling them to receive and transmit in both linear and circular co- and cross-polarizations. While single dipoles can also polarize incident waves, their behavior is highly contingent upon the dipole orientation, as demonstrated in [31]. In contrast, the crossed dipole offers advantages due to its two axes, resulting in two characteristic doughnut-shaped scattering patterns, respectively. This configuration ensures broader coverage and more robust polarization. Nevertheless, it is crucial to acknowledge the persistence of orientation-dependent scattering behavior, requiring careful consideration. The general tracer design can be seen in Figure 12.

Here, a crossed dipole is formed by two L-shaped dipoles. The sphere encompassing the dipoles is constructed from a high dielectric material, strategically chosen to electrically shorten the dipoles and enable their operation at lower frequencies. For this instance, a relative permittivity of $\varepsilon_{r}=6$ was selected. Additionally, the dipoles should be crafted from a highly electrically conducting material. It is essential to highlight that the dipoles maintain a separation, with a slight gap introducing an offset in the axes of each straight dipole. This intentional misalignment contributes to favorable cross-polarization behavior, as perfectly aligned dipole axes tend to yield weaker cross-polarization. The sphere and dipole possess diameters of 20 mm and 1.5 mm, respectively. The dipole arms measure 16 mm in length, resulting in a dipole resonance frequency of approximately 3.8 GHz. This resonance is observable in the co-polarization by the circle of strong scattering at 3.8 GHz and in the cross-polarization by two half-circles of strong scattering at the same frequency, as illustrated in Figure 13.

Importantly, the tracer’s frequency feature is suitably positioned within the low clutter region of the 10 mm bulk particles in the cross-polarization channels. For simulation purposes, the model depicted in Figure 12a, featuring the crossed dipole orientation, was employed. Furthermore, Figure 13 provides a comparison of the scattering characteristics with a 20 mm metal sphere utilized in previous work. The sphere under consideration does not exhibit any distinctive frequency-dependent behavior or heightened scattering characteristics that facilitate its detection. Nevertheless, the sphere can be effectively detected in randomized beds through radar measurements, as demonstrated in [20]. This capability stems from the radar approach, which permits spatial or temporal resolution of the measurement. Such resolution techniques are commonly employed for the precise localization of radar targets. In this context, clutter from the packed bed is staggered in the time domain, potentially enabling the detection and localization of the metal sphere. To achieve this, the radar system must possess sufficient bandwidth to ensure adequate time resolution, and the sphere must exhibit a robust scattering behavior. The utilization of the time domain for clutter and tracer separation is a fundamental aspect of the signal-processing approach in this work. Consequently, the range-frequency behavior of the crossed dipole is illustrated in Figure 14 for both polarizations.

This representation is achieved through the short-time Fourier transform (STFT), incorporating the relationship between distance and time, R=t·c/2, originating from the monostatic radar approach. The STFT inherently involves a trade-off between temporal and frequency resolution. The temporal resolution $r_{t}=\frac{B}{2c}$, which simultaneously defines the range resolution, relies on the measurement bandwidth, *B*, and the constant for the speed of light, c. The frequency resolution on the other hand is dictated by the length of the transformation window. Using shorter windows allows for accurate representation of rapid changes in the frequency domain, though at the expense of compromised range resolution. In this work, a window size of $w_{f}=\frac{B}{5}$ was selected for the corresponding frequency window length. In the simulation results of Figure 14, the tracer was positioned at a distance of 0.67 m from two dual-polarized antennas separated by a bistatic angle of 90°. The antenna reflection behavior was computed from the results obtained through a simulation involving only the antennas. However, the path length offsets the absolute distance to approximately 0.88 m. In strong alignment with Figure 13a,b, the range-frequency depiction illustrates the dipole resonance behavior in both polarizations.

## 4. Setup and Methods

### 4.1. Measurement Setup

The measurement setup was adapted from Figure 1 and can be seen in Figure 15 in an anechoic chamber.

The design of the measurement cell or reactor draws inspiration from industrial grate systems, commonly characterized by cuboid configurations. Constructed with 1 cm-thick poly(methyl methacrylate) (PMMA), and exhibiting a relative permittivity of $\varepsilon_{r}=2.6$ [32], the reactor’s inner dimensions measure 300 mm in width, 300 mm in length, and 450 mm in height. Consequently, the measurement volume encompasses 0.0405 m^3^ or 40.5 L. In the context of bulk material, spherical polyoxymethylene (POM) spheres with a diameter of 10 mm were employed. This results in an approximate packing of 50,000 spheres within the measurement system. The relative permittivity of POM typically falls within the range of $\varepsilon_{r}=2.87\ldots 3.48$ [33]. For the specific spheres utilized in this study, the determined relative permittivity was $\varepsilon_{r}=3.2$ [31]. Unlike industrial systems, the utilized reactor lacked any feeding or discharge mechanisms, and manual intervention was employed for sphere mixing. Positioned around the grate system were four circularly mounted dual-polarized antennas (QEH20E—RFspin s.r.o., Prague, Czech Republic) affixed to a polyamide-6 (PA-6) framework. All antennas possess a cross-polarization isolation better than 33 dB in the monostatic case and better than 44 dB for all bistatic cases. This was proven by preceding measurements of a metallic sphere with a 17 cm diameter in the middle of the setup without the reactor. For reactor measurements, the horizontal antennas were aligned with the middle height, while the vertical antennas were positioned along the middle width of each reactor sidewall. To comply with far-field conditions, their distance from the reactor was set at 51 cm. The antennas were linked to a calibrated vector network analyzer (VNA) from Rohde & Schwarz (ZNB 8—Rohde & Schwarz, Munich, Bavaria, Germany), conducting measurements based on the FMCW radar principle. To connect all eight antenna ports (four antennas with two polarizations), a switching matrix was utilized, calibrated with a corresponding calibration matrix (ZN-Z154—Rohde & Schwarz, Munich, Bavaria, Germany) using match, open, short, and through calibration standards. The system bandwidth was configured from 1.5 to 8.5 GHz with 1001 sweep points and an IF bandwidth of 1 kHz, leading to prolonged measurement times due to the serial measurement procedure of the switching matrix. While this may be insufficient for real-time measurements, the focus of this work is on validating the general measurement principle and ensuring optimal conditions. Future efforts will focus on adapting the measurement principle and replacing the time-multiplex mode with parallel measurements across all receiving channels. Given the substantial data generated by the four polarimetric antennas, we provide a detailed analysis of the measurement channels associated with antennas 1 and 2, presented here as an illustrative example, to validate the efficacy of target detection.

### 4.2. Characterization of Tracer and Bulk

As the tracer particle, a crossed dipole constructed from copper wires was encapsulated in an aluminum oxide potting compound with relative permittivity of $\varepsilon_{r}=6$ [31]. The dimensions of the sphere and dipole arms align with those employed in the simulations detailed in Section 3.3. To validate the manufacturing process, measurements were conducted on the individual tracer particle to extract its frequency features. For this purpose, the tracer was positioned on a bracket designed to minimize reflections.

To ensure precise alignment with the antennas, a cross-line laser was employed for tracer positioning, and it was subsequently removed from the measurements after adjustment. In all measurements, a free-space measurement without the tracer was recorded and subtracted to eliminate potential interference from the antennas and the bracket. The processed range-frequency behavior of the tracer in the orientation specified in Figure 16 (one dipole axis aligns with the y-axis and the other one with the z-axis of the coordinate system) is depicted in Figure 17a,b for co- and cross-polarizations, respectively.

The observed behavior of the tracer closely mirrors the results anticipated by simulations in Figure 14, affirming the intended behavior of the tracer. Furthermore, Figure 17c,d reveal the range-frequency scattering behavior of the bulk with 10 mm diameter spheres. The co-polarization clutter overshadows the tracer particle, while the cross-polarization feature of the tracer stands out amid the low-frequency cross-polarization clutter of the bulk. To facilitate additional comparison, Figure 17e,f present measurements of the bulk material with an embedded tracer. In this configuration, the reactor was initially filled halfway with the bulk material, followed by the precise placement of the tracer particle at the center using a positioning unit. Subsequently, the remaining volume of the reactor was filled with the bulk medium, and the positioning device was removed. The presented measurements illustrate that in the co-polar orientation, the dipole is entirely overshadowed by the bulk material. Conversely, in the cross-polar orientation, a separation of the frequency-dependent characteristics of both the bulk material and the tracer particle is evident. Since the detection sensitivity of the crossed dipole varies with different orientations, the detectability of the tracer particle concerning its orientation to a bistatic antenna pair was estimated. Therefore, measurements in cross-polarized channels were conducted on the crossed dipole for several different orientations. To analyze the resonance behavior of the tracer, the magnitudes of the respective channels in the range-frequency domain, ranging from 3 to 5 GHz and at a distance of 0.83 to 0.89 m, were averaged. The results of the measurements showed that a comparison of the following cases is particularly interesting:(1)One dipole axis aligns with the y-axis, and the other one with the z-axis of the coordinate system shown in Figure 16.(2)A 45° rotation is performed on both the x-axis and the y-axis in relation to the orientation of (1).(3)A 90° rotation is performed on the x-axis and a 45° rotation on the y-axis in relation to the orientation of (1).

For these cases, the magnitudes varied in a range of 0.0102–0.0659. From these observations, a threshold of 0.03 could be derived, at which the tracer particle can be detected for a large number of different orientations due to a sufficient SNR. In all measurements, a free-space measurement without the tracer was recorded and subtracted to eliminate potential interference from the antennas and the bracket.

To further characterize the bulk, a corner reflector was strategically positioned between antenna 3 and the reactor, as illustrated in Figure 18. In this arrangement, three monostatic measurements of the empty and filled reactors were conducted with antenna 1, respectively. Utilizing spheres with a diameter of 10 mm, they were stirred between consecutive measurements to address the inherent inhomogeneity of the bulk material. The outcomes of the three subsequent measurements were averaged and subsequently transformed into the time domain using an inverse fast Fourier transformation (IFFT). Figure 19 displays the resulting time-domain scattering behavior. Reflections from the back wall and the reflector exhibit distinct propagation times, indicating an effective permittivity behavior of the bulk. While defining an effective permittivity is typically achieved through dielectric mixing models for small homogeneous inclusions within a matrix medium [34], this assumption cannot be universally applied to all frequencies due to the electrical large size of the spheres. Nevertheless, a comparable behavior is observed. Consequently, an apparent permittivity can be calculated from the time difference between the empty and filled reactor measurements by
(11)$$\varepsilon_{r,\mathrm{bulk}}={\left(\frac{\Delta t\cdot c}{2R}\right)}^{2}.$$ In (Equation 11), it is assumed that no alterations occur in the propagation path from antenna 1 to the reactor and from the reactor to the reflector. This assumption is deemed appropriate for the conducted measurements, given the robustness of the setup. Consequently, the calculated apparent permittivity is $\varepsilon_{r,\mathrm{bulk}}=2.03$, aligning well with the reported value in [20].

### 4.3. Algorithm

As illustrated in Section 3, the utilization of exclusively cross-polarized channels emerges as a logical choice for tracer particle detection, owing to the minimal clutter inherent in the bulk environment. The setup configuration entails dual-sided measurements, with data acquired from two dual-polarized antennas, designated as Antenna 1 and Antenna 2. Accordingly, the measurement vector is expressed as follows: (12)$$k_{M}=\left[\begin{array}{c} S_{\mathrm{HV},21}\\ S_{\mathrm{HV},12}\\ S_{\mathrm{VH},21}\\ S_{\mathrm{VH},12} \end{array}\right].$$ The procedures necessary for detecting the crossed dipole in the bulk are outlined schematically in Figure 20 and subsequently elaborated in detail.

The initial requisite step in the detection process involves the transformation of the measurement vector $k_{M}$ to the range-frequency domain. The resulting representation allows for a detailed analysis of the signals in terms of their range-dependent frequency characteristics.

After the transformation, the algorithm advances to filter the particle resonance frequency components of the transformed signals. This filtering operation is crucial for isolating the relevant information of the crossed dipole features within the polarimetric data while mitigating the impact of noise and higher frequency clutter of the bulk. Simultaneously, range-gating is carried out to analyze the frequency components within a given range-resolution cell. Given the frequency-filtered range-cells $K_{M,\mathrm{Gated}}$, the analysis extends to second-order statistics, characterized by the eigendecomposition of the correlation matrix (see Section 2).
(13)$$\begin{array}{c} \begin{array}{cc} \left[T_{M}\right] & =K_{M,\mathrm{Gated}}\cdot K_{M,\mathrm{Gated}}^{*}=\left[V_{M}\right]\left[\Lambda_{M}\right]{\left[V_{M}\right]}^{-1}\\ \; & =\left[\begin{array}{ccc} v_{M,1} & v_{M,2} & v_{M,3} \end{array}\right]\left[\begin{array}{ccc} \lambda_{M,1} & 0 & 0\\ 0 & \lambda_{M,2} & 0\\ 0 & 0 & \lambda_{M,3} \end{array}\right]\left[\begin{array}{c} v_{M,1}\\ v_{M,2}\\ v_{M,3} \end{array}\right]. \end{array} \end{array}$$
where $\lambda_{M,1}>\lambda_{M,2}>\lambda_{M,3}>0$ indicate the intensities of the scattering mechanisms. Here, the eigenvector corresponding to the largest eigenvalue offers valuable insights into the scattering properties of the observed objects.

In the final stage of the algorithm, the weighted eigenvector $v_{M}=\lambda_{M,1}v_{M,1}$ for each measured range cell is systematically compared with the corresponding weighted eigenvector of the crossed dipole $v_{\mathrm{Ref}}=\lambda_{\mathrm{Ref},1}v_{\mathrm{Ref},1}$ through the inner product [26].
(14)$$\begin{array}{c} \langle v_{M},\;v_{\mathrm{Ref}}\rangle =\sum_{r}^{R}\lambda_{M,1}v_{M,1}\cdot \lambda_{\mathrm{Ref},1}v_{\mathrm{Ref},1} \end{array}$$
The eigenvector of the crossed dipole is determined through the eigendecomposition of the range cell, given by the gated STFT signal, denoted as $K_{\mathrm{Ref},\mathrm{Gated}}$. It is crucial to note that the eigenvector of the crossed dipole is dependent on its orientation with respect to the antennas. To establish a comparable reference for various dipole orientations, a database of eigenvectors corresponding to different orientations is needed. Thus, the inner product of the measured eigenvector and all reference eigenvectors in the database has to be calculated. The inner product serves as a quantitative measure of the polarization alignment between the two most dominant eigenvalue-weighted eigenvectors.

## 5. Measurement Procedure and Results

Leveraging the measurement configuration outlined in Section 4.1 and employing the algorithm detailed in Section 4.3, measurements were conducted on the crossed dipole embedded within the bulk medium. To trace the target within the bulk, the crossed dipole was systematically positioned at various locations, with its orientation corresponding to case (1) described in Section 4.2. Between consecutive measurements, the particles were stirred to account for the inherent inhomogeneity of the bulk material. The various positions of the crossed dipole during distinct measurements are depicted in Figure 15. As bistatic measurements are used for detection, this results in changes to the range profiles compared to the monostatic case. The corresponding measurement paths for points $P_{1}$, $P_{2}$, and $P_{3}$ are shown in Figure 21. Here, the propagation paths corresponding to the antenna lengths are neglected. Table 1 provides an overview of the monostatic and bistatic measurement paths.

As can be seen from the overview, there is an effective reduction in the distance traveled between two reactor walls for the monostatic case $|\Delta R_{13,\mathrm{mono}}\left|=\;\right|R_{1,\mathrm{mono}}-R_{3,\mathrm{mono}}|=\;56$ cm and the bistatic case $|\Delta R_{13,\mathrm{bi}}|=\;|R_{1,\mathrm{bi}}-R_{3,\mathrm{bi}}|=\;28$ cm. Due to this context, a reduction in the bistatic measured ranges of the crossed dipole can, therefore, be compared to a monostatic setup. Subsequently, the outcomes of the detection process are presented in Figure 22.

The trajectory of the tracer is discernible in the data plots. Nevertheless, an offset between point $P_{1}$ to the center point, $P_{2}$, and point $P_{3}$ to the center point, $P_{2}$, is evident. This displacement offset arises from the propagation characteristics of electromagnetic waves within the bulk with its apparent permittivity of $\varepsilon_{r,\mathrm{bulk}}=2.03$. Consequently, a non-linear relationship emerges due to the reduced propagation velocity of electromagnetic waves in the medium. This non-linear relationship introduces an offset relative to the particle’s linear displacement within the bulk. As the tracer progressively distances itself from the antennas and the emitted signals propagate within the bulk, the detected distance offset amplifies. Hence, the detected range of the tracer for point $P_{3}$ is situated farther from the center point $P_{2}$ than the distance of the tracer from point $P_{1}$ to the center point $P_{2}$. Considering the propagation within the medium, Table 1 presents the theoretically calculated distance values for the tracer positions. This calculation involves trigonometric considerations for the path length inside the medium, incorporating the determined apparent permittivity. In this context, the distance difference between $P_{1}$ and $P_{3}$ for the bistatic case is $|\Delta R_{13,\mathrm{bi}}\left|=\right|R_{1,\mathrm{bi}}-R_{3,\mathrm{bi}}|=40.1$ cm, which equals 20.05 cm in the radar range domain, as explained in Section 3.3. A comparison of this result with the distance of the peak values for $P_{1}$ and $P_{3}$ in Figure 22 reveals a very good agreement with an error of 1.5 %. For $P_{1}$ and $P_{2}$, the calculated distance difference is $|\Delta R_{12,\mathrm{bi}}|=|R_{1,\mathrm{bi}}-R_{2,\mathrm{bi}}|=18.5$ cm, corresponding to 9.25 cm in the radar considerations. However, Figure 22 shows a difference of 7.35 cm, approximately 20 %. Potential causes for this discrepancy include the diminished range resolution due to the STFT and the uncertainty principle. Consequently, the presence of coincidental strong clutter in close proximity to the marker may lead to a shift in the center of gravity while seeking the frequency features of the tracer. Additionally, the exact position of the tracer cannot be verified by other methods and remains unknown. Small displacements may occur due to falling particles after placing the tracer in the reactor and removing the positioning unit. Finally, the variance in radar range estimation is influenced by the successful separation of the tracer from the surrounding cluttering bulk particles. By recognizing bulk particles as a source of noise, the Cramér–Rao bound can be applied to determine the theoretical maximum precision of the measurements conducted. Referring to [35], the Cramér–Rao lower bound for range determination is as follows: (15)$$\mathrm{CRLB}\left(R\right)=\frac{3c^{2}}{\mathrm{SNR}\cdot N\cdot {\left(2\pi \cdot B_{\mathrm{eff}}\right)}^{2}}.$$ In (Equation 15), $B_{\mathrm{eff}}$ is the effective used bandwidth, which—for the STFT processing—corresponds to the chosen windows size $w_{f}$, and *N* is the number of measurements. Based on this, Figure 23 provides the achieved root-mean-square error (RMSE) for different effective bandwidths and SNRs for single measurements. For the conducted measurements, SNRs of 12 to 15 dB were achieved (see, e.g., Figure 17d,f) and an effective bandwidth of $B_{\mathrm{eff}}=B/5$ was used. This results in an estimated range error greater than one centimeter, which adds to the causes of discrepancy in range determination, but is still lower than the achieved error.

Thus, the overall measurement error for the position of the tracer in the middle of the reactor should be interpreted by taking into account all of the sources of errors mentioned before. The key takeaway from the results is the successful extraction of the frequency-dependent features associated with the cross-polarizing particle. This accomplishment signifies the approach’s capability to isolate the target’s scattering mechanisms from the surrounding clutter of the bulk. The ability to distinguish and extract the signal of interest from the environmental clutter enhances the overall effectiveness of the polarimetric radar system in detecting and tracking cross-polarizing tracer particles within cluttered environments.

## 6. Conclusions

This study introduces a novel measurement approach for detecting tracers within dynamic particle assemblies, employing polarimetric FMCW radar measurements. This approach effectively isolates the cross-polarizing scattering characteristics of the tracer from the complex and inhomogeneous cluttering environment. Leveraging the weaker cross-polarization clutter, compared to the co-polarization case, enables more efficient filtering. Our investigation includes a detailed analysis of the frequency-dependent behavior of cross-polarization clutter through extensive simulation studies involving different bulk configurations. From these findings, we propose a general design strategy for tracer particles, employing crossed dipoles embedded in high dielectric materials, such as ceramics, to shift their resonance frequency and polarization mechanism to lower frequency regions. This strategic design allows tracers to stand out against the bulk clutter, facilitating their detection. The scalable nature of our approach accommodates various bulk configurations, enabling tracer particles to match the size of surrounding bulk particles. This similarity in size ensures comparable dynamic behavior between the tracer and bulk particles, leading to more insightful conclusions about particle processes. Furthermore, our approach proves applicable to scenarios involving strongly varying or moving bulk configurations, eliminating the need for a reference measurement subtraction. Additionally, we introduce an algorithm for detection based on correlated eigenvectors of polarimetric range-frequency signatures, successfully validating our proposed approach by measuring the movement of a tracer within a dynamic bulk.

While our study represents a significant advancement in tracer detection and scalability, it is essential to acknowledge its limitations. Balancing identification and localization in the range-frequency domain involves trade-offs between resolution in both domains. Achieving accurate range detection necessitates spatial resolution for distinguishing tracers from clutter, while sufficient frequency resolution is crucial for detecting resonant tracers with limited bandwidths. Challenges arise when placing the frequency feature of the tracer in very low-frequency regions or when dealing with small tracers relative to the wavelength of the feature frequency. A potential solution involves employing higher measurement bandwidths, though practical constraints may limit this approach. These challenges present opportunities for innovative tracer designs and improved detection algorithms in future research. The study also lays the groundwork for exploring tracer rotation detection using the orientation-dependent scattering characteristics of crossed dipoles. Alternatively, enhancing tracer coverage with omnidirectional cross-polarization characteristics is another avenue for future investigation. Finally, additional processing techniques such as radar imaging can be directly applied to the presented data and will be employed in future work.

In conclusion, our work extends the applicability of non-intrusive particle localization and tracking using radar systems, demonstrating its potential to make significant contributions across various research fields dealing with particulate systems. 

## Figures and Tables

**Figure 1 sensors-24-02673-f001:**
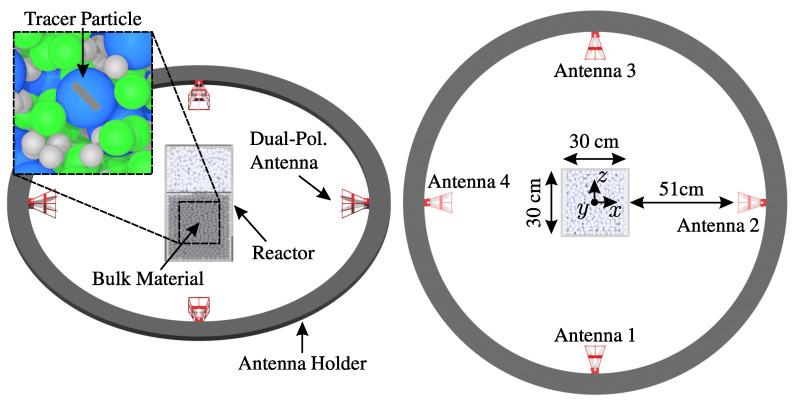
Measurement setup for the tracking of tracer particles in granular bulks.

**Figure 2 sensors-24-02673-f002:**
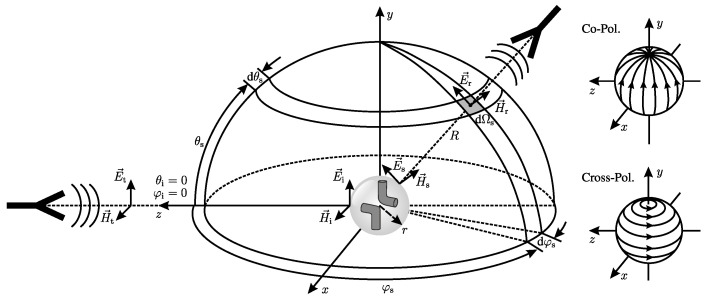
Schematic for the definition of the radar’s cross-section and the polarization basis.

**Figure 3 sensors-24-02673-f003:**
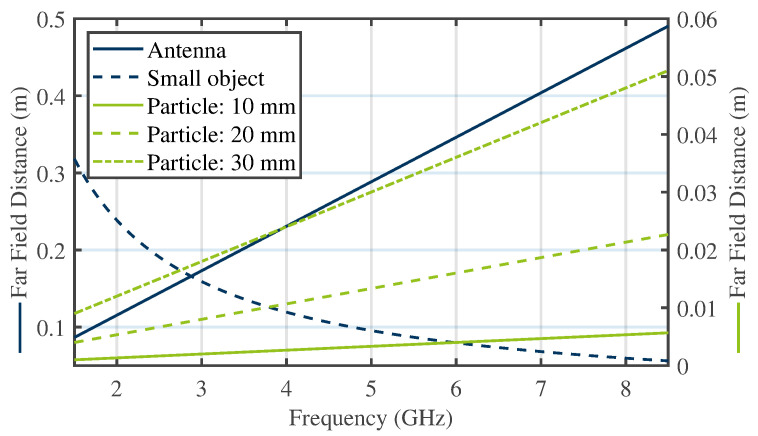
Far-field distances for the antenna and particle sizes.

**Figure 4 sensors-24-02673-f004:**
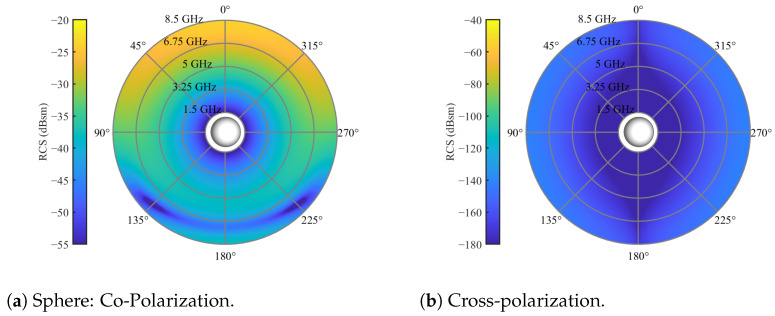
RCS of a single spherical particle with a diameter of 20 mm and relative permittivity of $\varepsilon_{r}=3.2$ for an elevation angle of $\theta_{s}=0$ and full azimuth orbit.

**Figure 5 sensors-24-02673-f005:**
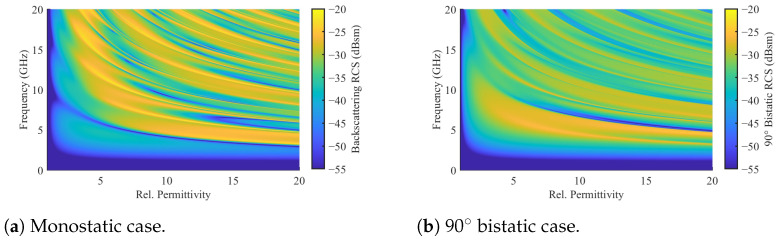
Permittivity dependency of the scattering of a dielectric sphere in the co-polar monostatic and the 90° bistatic case based on the Mie scattering theory [28].

**Figure 6 sensors-24-02673-f006:**
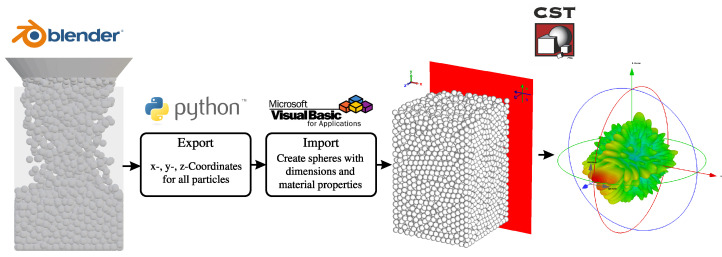
Co-simulation framework for the generation and electromagnetic characterization of packed particle beds.

**Figure 7 sensors-24-02673-f007:**
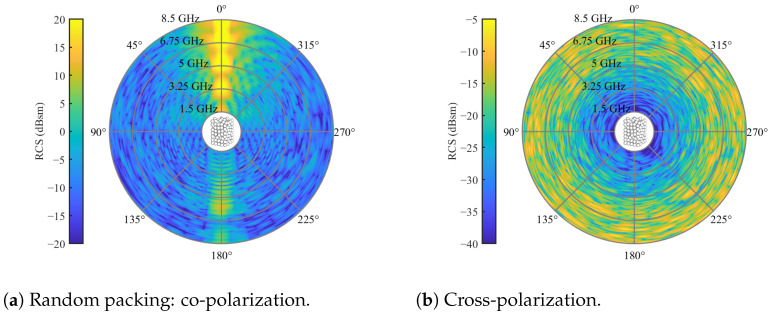
RCS of spherical particles with a diameter of 20 mm and relative permittivity of $\varepsilon_{r}=3.2$ in a random packing for an elevation angle of $\theta_{s}=0$ and full azimuth orbit.

**Figure 8 sensors-24-02673-f008:**
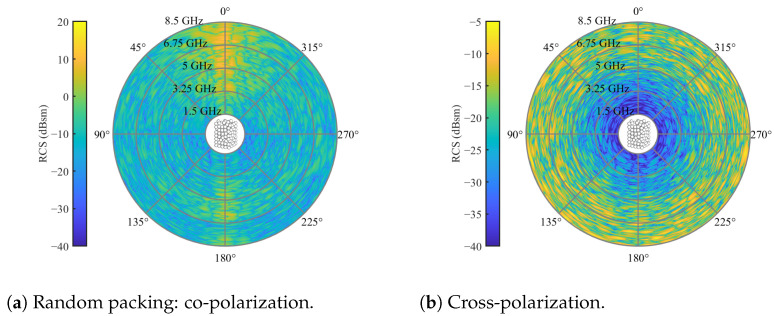
Subtracted RCS of two randomly packed particle beds with a sphere diameter of 20 mm and relative permittivity of $\varepsilon_{r}=3.2$ for an elevation angle of $\theta_{s}=0$ and full azimuth orbit.

**Figure 9 sensors-24-02673-f009:**
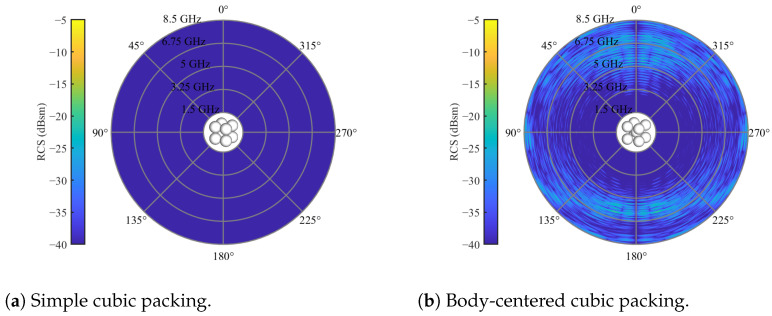
Cross-polarized RCS of spherical particles with a diameter of 20 mm and relative permittivity of $\varepsilon_{r}=3.2$ in different packings for an elevation angle of $\theta_{s}=0$ and full azimuth orbit.

**Figure 10 sensors-24-02673-f010:**
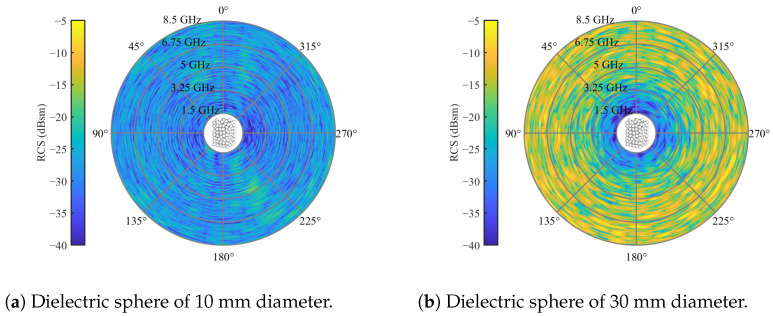
Cross-polarized RCSs of spherical particles with diameters of 10 and 30 mm and relative permittivity of $\varepsilon_{r}=3.2$ in random packing for an elevation angle $\theta_{s}=0$ and full azimuth orbit.

**Figure 11 sensors-24-02673-f011:**
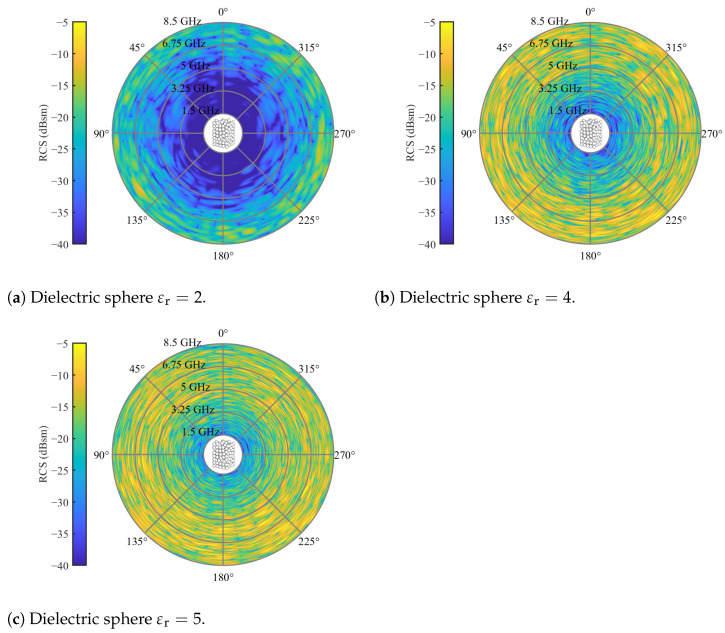
Cross-polarized RCS of spherical packed particles with different relative permittivities of $\varepsilon_{r}=\left[2\;4\;5\right]$ for an elevation angle of $\theta_{s}=0$ and full azimuth orbit.

**Figure 12 sensors-24-02673-f012:**
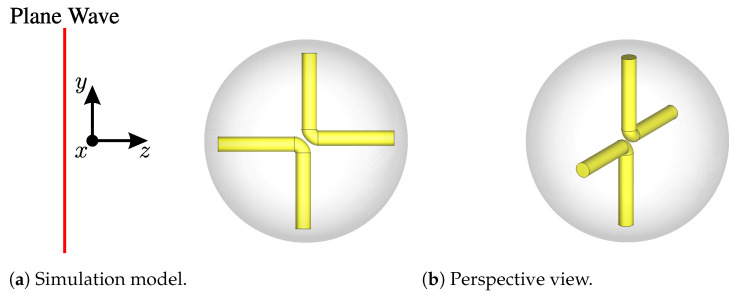
Tracer particle: Crossed dipole embedded in the dielectric sphere.

**Figure 13 sensors-24-02673-f013:**
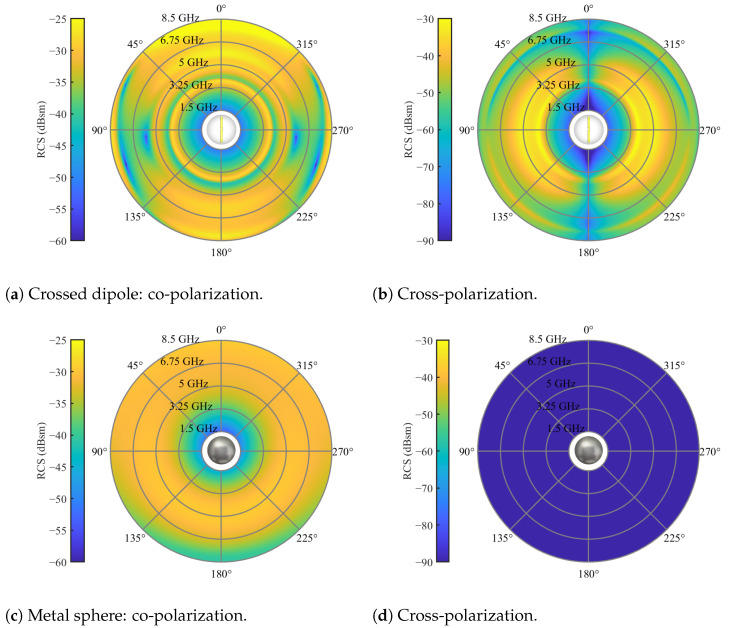
RCS of tracer particles for an elevation angle of $\theta_{s}=0$ and full azimuth orbit.

**Figure 14 sensors-24-02673-f014:**
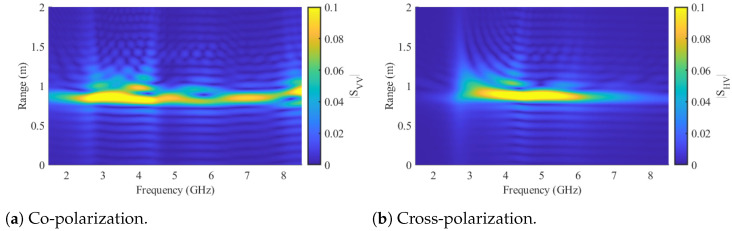
Range frequency representation of the simulated crossed dipole tracer particle.

**Figure 15 sensors-24-02673-f015:**
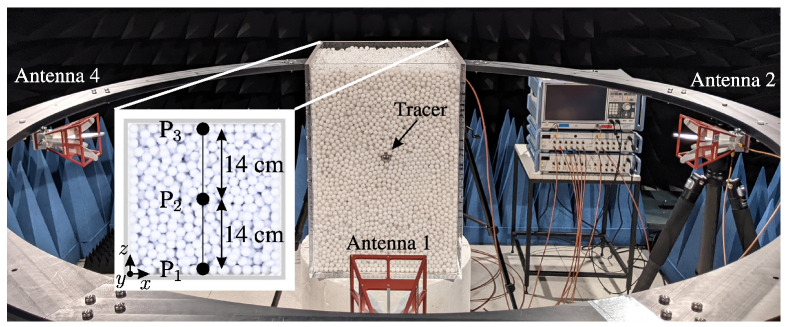
Measurement configuration in an anechoic chamber for the tracking of movements of tracer particles in granular bulks.

**Figure 16 sensors-24-02673-f016:**
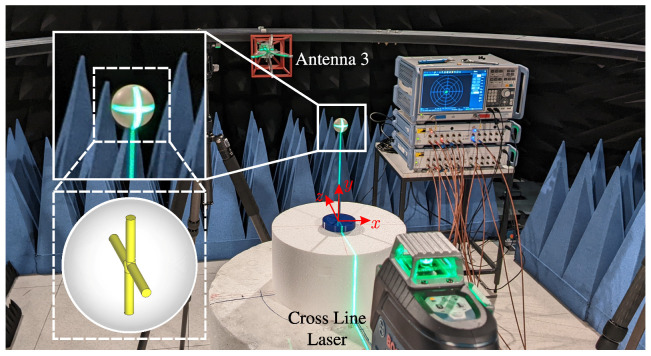
Measurement configuration to extract the range-frequency features of the tracer particle.

**Figure 17 sensors-24-02673-f017:**
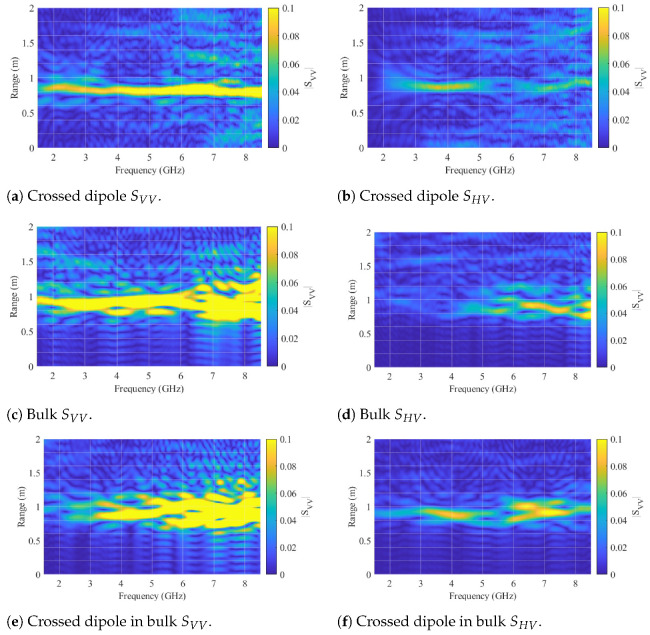
Range-Frequency representation of the manufactured crossed dipole tracer particle and a particle-filled reactor (10 mm diameter spheres) without a tracer.

**Figure 18 sensors-24-02673-f018:**
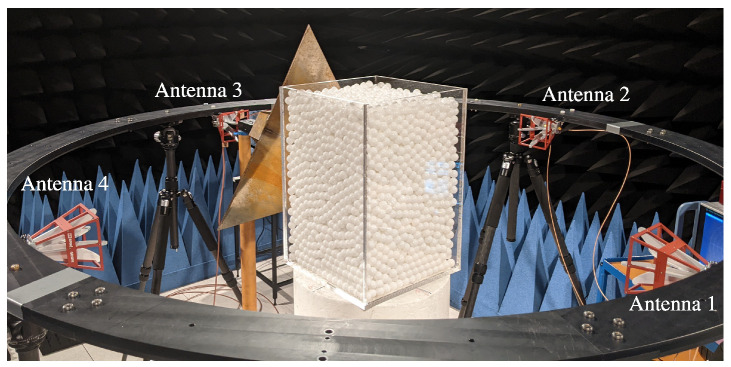
Measurement setup for the determination of the apparent relative permittivity of the bulk.

**Figure 19 sensors-24-02673-f019:**
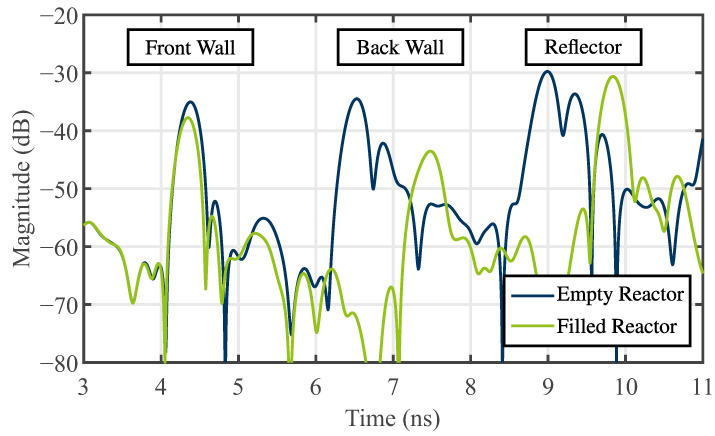
Time domain scattering for the determination of the apparent relative permittivity.

**Figure 20 sensors-24-02673-f020:**
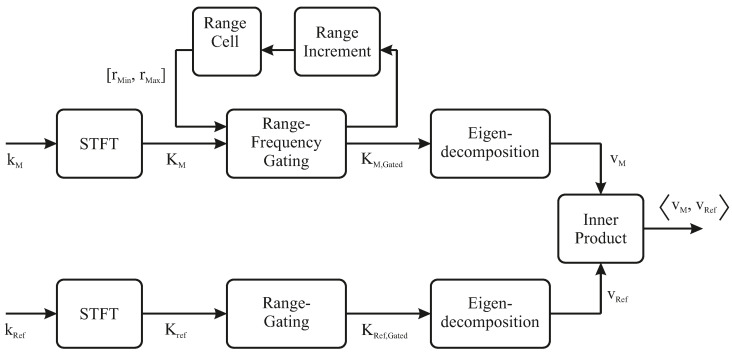
Flowchart for crossed dipole detection within non-homogeneous bulk systems.

**Figure 21 sensors-24-02673-f021:**
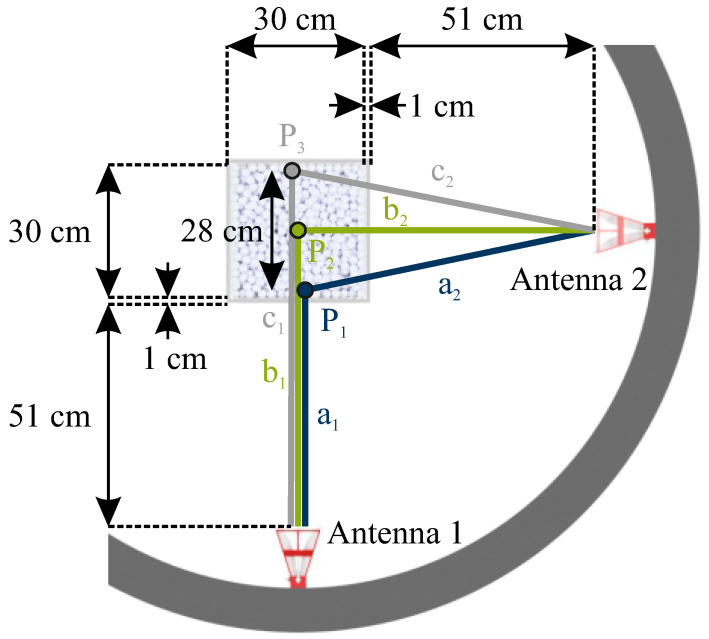
Schematic of measurement paths for the crossed dipole located at different positions within the bulk.

**Figure 22 sensors-24-02673-f022:**
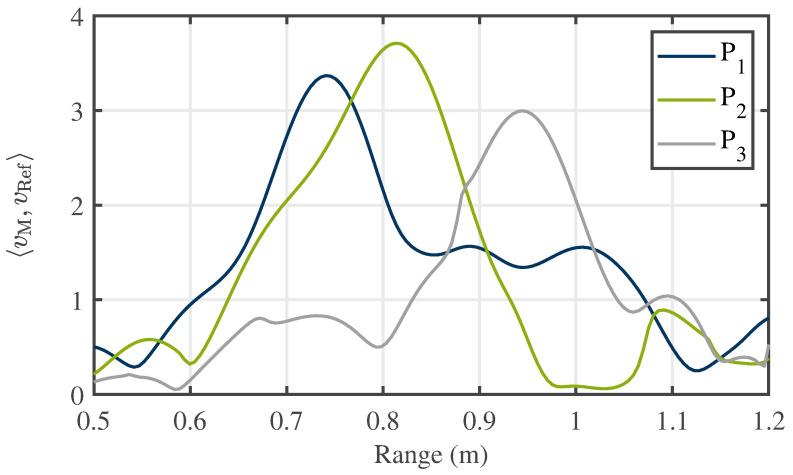
Extracted range profiles of systematically positioned crossed dipoles at different locations within the bulk.

**Figure 23 sensors-24-02673-f023:**
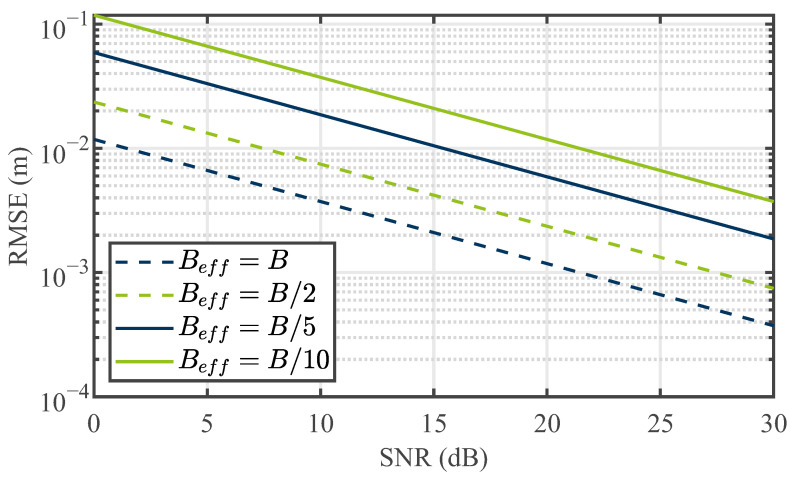
The RMSE of the positioning for different STFT window functions.

**Table 1 sensors-24-02673-t001:** Comparison of monostatic and bistatic propagation distances for different target positions with (w/) and without (w/o) bulk particles.

Target Position	Monostatic Range	Bistatic Range
$P_{1}$ w/o bulk	$R_{1,\mathrm{mono}}=2\cdot a_{1}=106$ cm	$R_{1,\mathrm{bi}}=a_{1}+\sqrt{{\left(b_{1}-a_{1}\right)}^{2}+b_{2}^{2}}=121.5$ cm
$P_{2}$ w/o bulk	$R_{2,\mathrm{mono}}=2\cdot b_{1}=134$ cm	$R_{2,\mathrm{bi}}=b_{1}+b_{2}=134$ cm
$P_{3}$ w/o bulk	$R_{3,\mathrm{mono}}=2\cdot c_{1}=162$ cm	$R_{3,\mathrm{bi}}=c_{1}+\sqrt{{\left(c_{1}-b_{1}\right)}^{2}+b_{2}^{2}}=149.5$ cm
$P_{1}$ w/ bulk	$R_{1,\mathrm{mono}}=106$ cm	$R_{1,\mathrm{bi}}=127.6$ cm
$P_{2}$ w/ bulk	$R_{2,\mathrm{mono}}=146.1$ cm	$R_{2,\mathrm{bi}}=146.1$ cm
$P_{3}$ w/ bulk	$R_{3,\mathrm{mono}}=186.2$ cm	$R_{3,\mathrm{bi}}=167.7$ cm

## Data Availability

Data are contained within the article.

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
