# Peer review of "Enhanced Tracer Particle Detection in Dynamic Bulk Systems Based on Polarimetric Radar Signature Correlation"

_sensors, 2024, doi:10.3390/s24092673_

Round 1
Reviewer 1 Report
Comments and Suggestions for Authors
The measurement results are worth to be published.
Nevertheless, the overall paper is too long with respect to the content. Some lengthy introductory parts might be either regarded as superfluous or too educative for a journal paper (i.e. the fundamental discussion for the far field distance).
Fig. 13 is missing but still referenced in line 276. Maybe the two figures on top of page 12 should be Fig. 13?
Especially chapter 3 seems excessively long for its purpose. The RCS plots should be minimized to only the most important ones for this work.
In line 244 you say the resonance is observable in Fig. 14 but where can that be seen in Fig. 14?
Reviewer 2 Report
Comments and Suggestions for Authors
This is an interesting study and the authors have collected a unique dataset using a measurement system for detecting tracers within particle assemblies and polarimetric FMCW radar measurements. The proposed method is validated through comprehensive measurements, involving the systematic positioning of a tracer particle at various locations. There are some problems, which must be solved before it is considered for publication. If the following problems are well-addressed, this reviewer believes that the essential contribution of this paper are important for the detection of tracer particles within non-homogeneous bulk media.
1. The experimental setup in the article appears overly idealized and lacks practical application scenarios. It would be beneficial to consider real-world applications such as MRI, ECT, and X-ray imaging, which have specific practical uses. The study should aim to address practical applications to enhance its relevance.
2. It is essential to investigate whether other scholars have addressed similar issues highlighted in the paper and proposed alternative methods. Conducting a comparative analysis with existing research can provide valuable insights and contribute to the advancement of the field.
3. When detecting targets in a medium, referencing literature on polarimetric ground-penetrating radar, which has numerous practical applications, can offer valuable insights and enhance the study's applicability.
4. The article's structure places excessive emphasis on the experimental section. Consider relocating the systematic introduction from the Section 1. Introduction to later sections to improve the document's organization and aid reader comprehension.
5. The use of crossed dipoles as tracers is an intriguing proposition. It would be beneficial to clarify whether these crossed dipoles feature a unique design and if they represent the optimal choice for tracers in the given context.
6. Crossed dipoles are very sensitive to orientation. In real-world scenarios where objects are in motion, various orientations may occur. It is advisable to conduct statistical analyses to account for these variations and validate the tracking performance under dynamic conditions.
7. The article lacks a description of the positioning accuracy under different signal-to-noise ratios (SNRs). It is recommended to include measurements of background SNRs and conduct quantitative analyses to assess the localization precision.
8. In Section 2. Fundamentals, it is advisable to include references for the polarimetric expressions used. These expressions are established in the literature, and referencing relevant sources will provide credibility and context to the study.
9. The authors are advised to remove experimental sections unrelated to the main theme of the paper, such as the analysis of octahedrons and cubes, which are not further discussed anymore. These sections do not contribute significantly to the paper's central theme.
10. The residual amount of -5dB, mentioned in Line 234 on Page 7, should not be overlooked. Utilizing Monte Carlo simulations for quantitative analysis can provide a more comprehensive understanding of random scenarios.
11. Due to potential errors in CST's time-domain solver, particularly when simulating cross-polarization, it is recommended to address the magnitude of these errors. Validation using real systems at specific orientations and frequencies can verify the accuracy of the simulations.
12. Consider addressing coupling effects between different polarized antennas and channels in real systems to account for non-ideal scenarios. Testing with metallic spheres can help identify system errors related to cross-polarization in practical measurements.
13. Conducting control experiments is essential to demonstrate the effectiveness of using crossed dipoles. Consider replacing crossed dipoles with metallic spheres for comparative experiments to validate the proposed methodology.
14. The discrepancy in signal amplitude between the metal corner reflector and the dielectric wall in Figure 20 is only around 5dB, which deviates from conventional expectations. Clarifying this discrepancy can enhance the paper's credibility.

What is more, the structure of the paper is a bit confusing and the format is not rigorous. Please make detailed modifications according to the journal’s template. the authors may need edit the manuscript carefully and pay particular attention to structure and format.
15. In Table 1, the abbreviation "w/o" should be clarified to provide a clear understanding of its meaning within the context of the table.
16. Including a summary at the end of each section would enhance coherence and provide a structured flow of information throughout the paper.
17. Some paragraphs start with an indentation, while others start without it, and the length of the initial space in each paragraph varies.
18. Figure 13 lacks a caption.
19. The positioning of some images is distant from the corresponding descriptions in the text, requiring readers to flip back and forth to view them.
20. The order of Figures 12 to 14 in the text description is disorganized; it is recommended to align them with the order of the paragraphs.
21. The formatting is inconsistent and may hinder readability; for example, Line 256 on Page 9 stands alone, appearing really abrupt.
22. The document size is excessively large, over 120MB, which may impede download and readability. It is advisable to reduce the file size for easier access by readers.
Round 2
Reviewer 1 Report
Comments and Suggestions for Authors
Thanks for the revision and reply, which adresses all comments. The manuscript is acceptable for publication (even though I am still of the opinion that the introduction is too long, but that is no hinder for publication).
One minor thing might be considered for Figure 17 (Fig. 18 in V1):
I would recommend to use a kind of grid on these figures. It seems that the range measurement slightly depends on frequency (Figure 17a).
Reviewer 2 Report
Comments and Suggestions for Authors
Regarding Algorithm 4.3 and accompanying Equation (13) and (14), please clarify whether these are original contributions or derived from papers of other researchers. If they are your own methodologies, provide the necessary explication. Conversely, if they are grounded in prior research, kindly cite the corresponding references.
Comments on the Quality of English LanguageThe absence of initial spaces in certain paragraphs persists, as exemplified by Line 410 and 547. Kindly scrutinize the entire document for similar occurrences.
